# Robust Strength Behavior Modeling of Coarse-Grained Soils Using HSIC-Guided Stable Learning

## Abstract

Coarse-grained soils are widely employed in infrastructure construction, and capturing their strength behavior is vital for ensuring the structural integrity of engineering systems. In recent years, artificial intelligence (AI) techniques have shown significant promise in advancing investigations in this area. Nevertheless, conventional AI models often exhibit limited robustness when confronted with distributional shifts in the data. To tackle these limitations, this study introduces a stable learning framework based on the Hilbert-Schmidt Independence Criterion, referred to as HSIC-StableNet, for predicting deviatoric stress–axial strain ($q$–$\varepsilon_a$) curves that represent the strength characteristics of coarse-grained soils. The proposed method initially adopts HSIC with the exact kernel method to replace the F-norm combined with the approximate kernel method, strategically reweighting training samples to enhance the stable learning module and integrating it with a deep neural network. The experimental results indicate that HSIC-StableNet consistently surpasses conventional DNN models and a previously introduced stable learning approach, SNN, across key metrics such as R², MSE, MAE, and MAPE. Furthermore, the model demonstrates strong performance in estimating the strength behavior of coarse-grained soils with large particle sizes by utilizing data samples from soils with smaller particles. This capability contributes to alleviating the data scarcity challenge in geotechnical engineering, where acquiring adequate large-particle soil data through costly triaxial tests remains difficult.

## 1 Introduction

Coarse-grained soils, composed of over 50% particles larger than 0.075mm, are widely used in infrastructure due to their favorable engineering properties. Their strength behavior, often described by deviatoric stress–axial strain ($q$–$\varepsilon_a$) curves, is typically obtained through triaxial tests or discrete element modeling (DEM) (Yan S., 2022; Bai J., 2022; Chen J., 2023; Wang L., 2022; Ren S., 2025; H. et al., 2025; G. & S., 2000; Lin S., 2024). However, these methods are costly and time-consuming, limiting their practical scalability (Zhang X., 2023; Ovalle C., 2020; Yao Y., 2012; Kidane M., 2021).

Artificial intelligence (AI) has shown promise in predicting soil strength behavior (Pham B. T., 2018), but traditional machine learning (ML) assumes that training and test data share the same distribution. In reality, coarse-grained soil data are often sparse or imbalanced, especially for large particle sizes, leading to distribution shifts that degrade model generalization—a challenge known as out-of-distribution (OOD) generalization (Shen Z., 2020; Yu H., 2024; Arjovsky M., 2019).

Domain Generalization (DG) methods attempt to address OOD by learning representations across multiple source domains (Rahimian H., 2019; Creager E., 2021; Chen Y., 2022; Zhao Y., 2021), but their reliance on diverse, well-labeled datasets limits their practicality. To overcome this, stable learning has emerged as an effective strategy by reweighting training samples to reduce reliance on spurious correlations (Cui P., 2022). This is achieved through feature decorrelation, typically measured by the Hilbert-Schmidt Independence Criterion (HSIC). While HSIC is computationally intensive and often approximated in large-scale applications (Yao J., 2023), the moderate size of soil datasets allows for exact computation, enhancing prediction accuracy.

Moreover, most stable learning methods are developed for classification tasks, whereas this study focuses on regression. To fill this gap, we propose HSIC-StableNet, a novel stable learning framework for regression, aimed at robust prediction of coarse-grained soil strength. The key contributions of this work include:

1.**Incorporation of Exact Kernel-Based Dependency Measures into Stable Learning**: This study introduces the HSIC-StableNet framework, which utilizes the Hilbert-Schmidt Independence Criterion (HSIC) to perform precise sample reweighting aimed at reducing feature dependencies. In contrast to conventional stable learning approaches that rely on approximate kernel estimations, the use of exact kernel methods enhances both the predictive accuracy and robustness of the model, particularly in modeling the strength behavior of coarse-grained soils.

2.**Generalization of Stable Learning to Regression Problems**: While most existing stable learning methods are developed for classification tasks, this work extends the paradigm to regression scenarios by embedding a stable learning mechanism within a regression framework. This extension enables effective feature decorrelation and improved generalization in regression-based prediction tasks, thereby broadening the applicability of stable learning to complex engineering problems involving limited and noisy data.

3.**Alleviating Data Scarcity via Cross-Scale Learning Strategy**: To address the limited availability of triaxial test data for coarse-grained soils with large particle sizes, the proposed framework exploits information from smaller-particle soil samples to infer the strength behavior of larger-particle materials. This multi-scale learning approach effectively reduces the dependence on costly physical experiments and facilitates reliable predictions in data-scarce settings, offering a practical and economical solution for real-world geotechnical applications.

## 2 RELATED WORK

**Domain Generalization (DG).** DG aims to improve model robustness by learning representations that generalize to unseen domains. Existing methods mainly fall into two categories: (1) invariant feature learning, such as the entropy regularization approach by Zhao S. (2020), and (2) meta-learning, exemplified by Finn C. (2017), which simulates domain shifts via meta-training/testing splits. Despite their effectiveness, these methods often require domain labels, manual partitioning, and balanced sampling, limiting scalability in real-world applications (Zhou K., 2022; Wang J., 2022).

**Stable Learning.** Stable learning tackles out-of-distribution generalization by reweighting samples to reduce spurious correlations. Zhang X. (2021) proposed decorrelating causal and spurious features, while Ye W. (2024) introduced dependency-based weights to suppress unstable associations. Although effective in computer vision, these methods often rely on approximate kernel techniques with the Frobenius norm for efficiency, which may compromise accuracy and representation robustness.

## 3 METHODOLOGY

### 3.1 PROBLEM FORMULATION

This study aims to develop a data-driven model that predicts the deviatoric stress $q$ of coarse-grained soils under various triaxial test conditions. The task is formulated as a supervised regression problem, where the model learns the mapping from input features that represent test conditions and soil states to the corresponding deviatoric stress value. Each data sample is represented as a tuple $(\mathbf{x}, y)$, where $\mathbf{x} \in \mathbb{R}^n$ denotes the input feature vector, and $y \in \mathbb{R}$ represents the target output, i.e., the deviatoric stress $q$. The input vector $\mathbf{x}$ includes both test conditions and soil state parameters: $\mathbf{x} = [\sigma_3, d, h, d_{\max}, \rho_d, e, \mathrm{PSD}, \varepsilon_a]$, where $\sigma_3$ is the confining pressure; $d$ and $h$ are the container's diameter and height; $d_{\max}$ is the maximum particle size; $\rho_d$ is the dry density; $e$ is the void ratio; PSD denotes the particle size distribution curve; and $\varepsilon_a$ is the axial strain. The goal is to learn a function $f$ such that $y = f(\mathbf{x})$, enabling robust and accurate stress prediction across diverse soil conditions.

## 3.2 Overall Approach of HSIC-StableNet

To explore the relationship between deviatoric stress and axial strain in coarse-grained soils, we propose a novel stable learning framework, HSIC-StableNet, specifically designed to capture the intrinsic correlations between deviatoric stress ($q$) and axial strain ($q$–$\varepsilon_a$). The HSIC-StableNet architecture consists of two primary components: a deep neural network (DNN) and a stable learning module. The stable learning module aims to reduce statistical dependencies among input features within the DNN, thereby promoting feature independence and enhancing the overall learning process. It comprises two key submodules, namely sample reweighting and sample weight globalization, which function collaboratively to improve the model's robustness and generalization capabilities. The overall architecture of the proposed HSIC-StableNet is depicted in Figure 1.

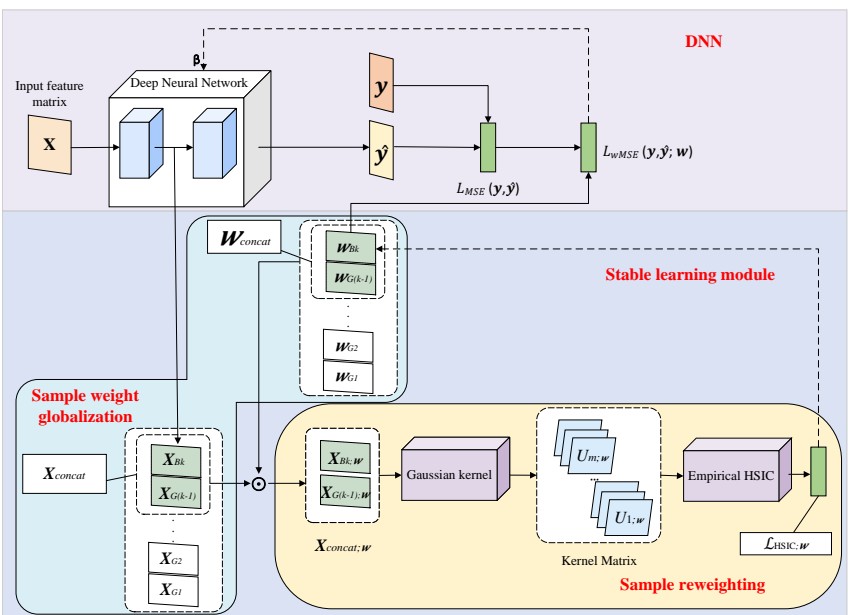

Figure 1: The model diagram of HSIC-StableNet.

## 3.3 Stable Learning

Stable learning utilizes exact kernel methods rather than approximate alternatives to enhance the generalization capability of the model. It generates sample weights through feature mapping and independence testing, effectively removing statistical dependencies among features to build a more robust model. The framework consists of two key components: a sample reweighting module and a sample weight globalization module, as shown in Figure 1.

### 3.3.1 Sample Reweighting with HSIC

To ensure effective feature decorrelation during sample reweighting, this module applies the Hilbert-Schmidt Independence Criterion (HSIC) to evaluate dependencies among input features. The features are first mapped into a Reproducing Kernel Hilbert Space (RKHS), after which HSIC is used to compute test statistics that quantify the degree of statistical dependence between feature pairs.

**(1) Feature Mapping**

To capture potential hidden dependencies among features, especially nonlinear relationships that are not evident in the original input space, this study adopts a feature mapping approach. Kernel methods are used to implicitly project data into a higher-dimensional space, which enables the learning of complex patterns through linear models. Instead of explicitly transforming the data, the method computes inner products between samples in the mapped space using a kernel function, as defined below:

$$K(x_i, x_j) = \langle \phi(x_i), \phi(x_j) \rangle \tag{1}$$

Here, $x_i$ and $x_j$ represent the $i$-th and $j$-th data samples in the original input space, and $\phi(x_i)$ is an implicit mapping function. Kernel methods are particularly suitable for small-scale datasets, such as the coarse-grained soils dataset used in this study, where computational efficiency and high precision are essential.

In this study, the Gaussian kernel, a positive definite kernel function, is chosen to map data samples into the RKHS due to its mathematically simple form. It is defined as:

$$K(x_i, x_j) = \exp\left(-\frac{\|x_i - x_j\|^2}{2\sigma^2}\right), \quad \sigma > 0 \tag{2}$$

where $\|x_i - x_j\|^2$ represents the squared Euclidean distance between data samples $x_i$ and $x_j$, and $\sigma$ is a positive parameter known as the *bandwidth*, which controls the width of the kernel.

In this work, the Gaussian kernel is utilized to construct a kernel matrix for each input feature. Specifically, let $X \in \mathbb{R}^{N \times D}$ denote the input feature matrix, where $N$ is the number of data samples and $D$ is the number of features. The resulting kernel matrix $U_m$ for the $m$-th feature is defined as:

$$U_m = \begin{bmatrix} U_{11} & U_{12} & \cdots & U_{1N} \\ U_{21} & U_{22} & \cdots & U_{2N} \\ \vdots & \vdots & \ddots & \vdots \\ U_{N1} & U_{N2} & \cdots & U_{NN} \end{bmatrix} \tag{3}$$

Here, $U_{ij} = K(X_{i,m}, X_{j,m}) = \exp\left(-\frac{\|X_{i,m} - X_{j,m}\|^2}{2\sigma^2}\right)$, where $K$ represents the Gaussian kernel function applied to the $m$-th feature, $X_{i,m}$ and $X_{j,m}$ denote the $m$-th feature of the $i$-th and $j$-th data samples, respectively. $U_{ij}$ represents the inner product between two data points $X_{i,m}$ and $X_{j,m}$ in the Reproducing Kernel Hilbert Space (RKHS). Accordingly, a set of kernel matrices $\{U_1, \ldots, U_m, \ldots, U_D\}$ is obtained, which will be utilized for removing correlations among features as discussed in the following.

**(2) Removing Dependencies among Features with HSIC**

Eliminating dependencies helps reduce spurious correlations by preventing the model from relying on coincidental or non-causal feature associations that may not hold across different datasets. This enhances the model's ability to focus on robust patterns, thereby improving generalization to unseen data.

In this study, the Hilbert-Schmidt Independence Criterion (HSIC) is employed to quantify dependencies among features, followed by minimizing HSIC values through sample reweighting to enhance feature independence.

**• Hilbert-Schmidt Independence Criterion(HSIC)**

**Definition 1** (Hilbert-Schmidt Independence Criterion) (Gretton A., 2005): Let $X \in \mathbb{R}^{N \times D}$ be the input matrix with $N$ samples and $D$ features. Suppose $X_{:,m}$ and $X_{:,n}$ are the vectors representing the $m$-th and $n$-th feature vectors across all samples, respectively. The cross-covariance operator between these two vectors is denoted as $C_{X_{:,m}, X_{:,n}}$. The HSIC between $X_{:,m}$ and $X_{:,n}$ is defined as:

$$\mathrm{HSIC}(X_{:,m}, X_{:,n}; \mathcal{F}, \mathcal{G}) = \|C_{X_{:,m}, X_{:,n}}\|_{\mathrm{HS}}^2 \tag{4}$$

According to Theorem 4 in Gretton et al. (Gretton A., 2005), the squared Hilbert-Schmidt norm of the cross-covariance operator $\|C_{X_{:,m}, X_{:,n}}\|_{\mathrm{HS}}^2$ is zero if and only if the features are statistically independent. This relationship is expressed in Equation (5).

$$\|C_{X_{:,m}, X_{:,n}}\|_{\mathrm{HS}}^2 = 0 \iff X_{:,m} \perp X_{:,n} \tag{5}$$

Empirical HSIC (Gretton A., 2005) provides a sample-based estimation of the Hilbert-Schmidt Independence Criterion, which quantifies the statistical dependence between two random variables using kernel methods. The empirical HSIC computes this dependence based on finite data samples, as shown in Equation (6):

$$\text{HSIC}(X_{:,m}, X_{:,n}) = \frac{1}{(N-1)^2} \text{tr}(U_m H U_n H) \tag{6}$$

where tr represents the trace of a matrix, $H$ is the centering matrix, $H = I - \frac{1}{N}\mathbf{1}\mathbf{1}^T$, where $I$ is the identity matrix and $\mathbf{1}$ is a column vector of ones. $U_m$ and $U_n$ are the kernel matrices (as defined in Equation (3)) corresponding to the feature vectors $X_{:,m}$ and $X_{:,n}$, respectively. Correspondingly, $U_m H$ can be obtained by Equation (7).

$$U_m H = \begin{bmatrix} U_{11} - \bar{U}_1 & U_{12} - \bar{U}_1 & \cdots & U_{1N} - \bar{U}_1 \\ U_{21} - \bar{U}_2 & U_{22} - \bar{U}_2 & \cdots & U_{2N} - \bar{U}_2 \\ \vdots & \vdots & \ddots & \vdots \\ U_{N1} - \bar{U}_N & U_{N2} - \bar{U}_N & \cdots & U_{NN} - \bar{U}_N \end{bmatrix} \tag{7}$$

where $\bar{U}_i = \frac{1}{N}\sum_{j=1}^{N} U_{ij}$. The computation of $U_n H$ follows a similar procedure.

• **Learning sample weights for feature decorrelation via HSIC-Loss**

As noted earlier, the closer the HSIC value between two feature vectors $X_{:,m}$ and $X_{:,n}$ approaches zero, the weaker their statistical dependence. In empirical HSIC, dependence is estimated using finite data samples, with each sample typically assigned equal weight. However, to effectively reduce this dependence for feature decorrelation, we initially propose optimizing the sample weights $\omega = [\omega_1, \omega_2, \ldots, \omega_N]$ to minimize the weighted HSIC, ideally driving it toward zero as expressed in Equation (8).

$$\text{HSIC}(\omega X_{:,m}, \omega X_{:,n}) = \frac{1}{(N-1)^2} \text{tr}(U_{m;\omega} H U_{n;\omega} H) \tag{8}$$

where $\omega \in \mathbb{R}^N$ represents the sample weights, while $U_{m;\omega}$ and $U_{n;\omega}$ denote the weighted kernel matrices corresponding to $X_{:,m}$ and $X_{:,n}$, respectively. In $U_{m;\omega}$, the standard kernel computation $U_{ij} = K(X_{i,m}, X_{j,m})$ is adjusted to $U_{ij} = K(\omega_i X_{i,m}, \omega_j X_{j,m})$, where $\omega_i$ and $\omega_j$ are the weights assigned to the $i$-th and $j$-th samples. The computation of $U_{n;\omega}$ follows in a similar manner.

To achieve independence between the *m*-th and *n*-th features, we minimize the HSIC value as defined in Equation (8). Accordingly, the HSIC values for all feature pairs are calculated, and Equation (9) is employed as the loss function to guide the optimization of sample weights.

$$\mathcal{L}_{\text{HSIC};\omega} = \sum_{m=1}^{N-1} \sum_{n=m+1}^{N} \text{HSIC}(\omega X_{:,m}, \omega X_{:,n}) \tag{9}$$

Theoretically, with an infinite sample size, it is possible to derive a set of weights that entirely eliminates feature dependence, resulting in $\mathcal{L}_{\text{HSIC};\omega} = 0$. In practice, however, given the finite dataset size, we minimize the sum of weighted HSIC values, as expressed in Equation (10).

$$\omega = \arg \min_{\omega} \mathcal{L}_{\text{HSIC};\omega} \tag{10}$$

In this study, we employ Mini-Batch Gradient Descent(MBGD) to minimize the objective function in Equation (9). By iteratively adjusting the parameters $\omega$ to minimize the objective function $\mathcal{L}_{\text{HSIC};\omega}$, feature correlations in the original dataset can be effectively reduced, enhancing model stability and generalization performance.

### 3.3.2 GLOBALIZING SAMPLE WEIGHTS

As mentioned earlier, MBGD is used to iteratively update sample weights. However, since MBGD processes only a subset of samples in each batch, the resulting weights remain localized, which can limit the effectiveness of reweighting in addressing statistical dependencies across the entire dataset. To address this issue, we propose a sample weight globalization module. This module aggregates and stores features and sample weights from previous batches. As depicted in Figure 1, the accumulated information is reloaded as global context, enabling comprehensive updates to sample weights throughout the entire dataset.

As shown in Figure 1, assume that the model has been trained with $k - 1$ batches of data. Let $X_{B1}, X_{B2}, \ldots, X_{Bi}, \ldots, X_{B(k-1)}$ represent the input feature matrices, where $X_{Bi}$ corresponds to the input features of the $i$-th batch (with $i$ denoting the batch number), capturing local information. Similarly, the accumulated global information after each batch is denoted as $X_{G1}, X_{G2}, \ldots, X_{Gi}, \ldots, X_{G(k-1)}$, where each $X_{Gi}$ represents the global context captured up to the $i$-th batch.

Additionally, let $\omega_{B1}, \omega_{B2}, \ldots, \omega_{Bi}, \ldots, \omega_{B(k-1)}$ represent the sample weights for the first $k - 1$ batches $X_{B1}, X_{B2}, \ldots, X_{Bi}, \ldots, X_{B(k-1)}$, all initialized to 1. The global sample weights corresponding to $X_{G1}, X_{G2}, \ldots, X_{Gi}, \ldots, X_{G(k-1)}$ are denoted as $\omega_{G1}, \omega_{G2}, \ldots, \omega_{Gi}, \ldots, \omega_{G(k-1)}$.

To allow samples from previous batches to contribute to the training of the current batch, we define $X_{\text{concat}}$ and $\omega_{\text{concat}}$ as concatenated input feature matrices and corresponding global sample weights, respectively. The concatenation is performed as follows:

$$X_{\text{concat}} = \begin{bmatrix} X_{G(k-1)}{}^T & X_{Bk}{}^T \end{bmatrix}^T \tag{11}$$

$$\omega_{\text{concat}} = \begin{bmatrix} \omega_{G(k-1)}{}^T & \omega_{Bk}{}^T \end{bmatrix}^T \tag{12}$$

$$\mathcal{L}_{\text{HSIC};\omega} = \sum_{m=1}^{N-1} \sum_{n=m+1}^{N} \text{HSIC}\left(\omega_{\text{concat}} X_{\text{concat}:,m}, \ \omega_{\text{concat}} X_{\text{concat}:,n}\right) \tag{13}$$

During the training process, $\omega_{G(k-1)}$ is kept fixed while $\omega_{Bk}$ is updated using the modified loss function $\mathcal{L}_{\text{HSIC};\omega}$, defined in Equation (13) as an updated version of Equation (9). $X_{\text{concat}:,m}$ denotes the $m$-th feature of concatenated matrix $X_{\text{concat}}$. Once the training reaches the maximum number of iterations, $\omega_{Bk}$ is obtained. We then fuse the global information $(X_{G(k-1)}, \omega_{G(k-1)})$ with the local information $(X_{Bk}, \omega_{Bk})$ using Equations (14) and (15). This process effectively incorporates information from all previous batches to optimize the current sample weights.

$$X_{Gk} = \alpha X_{G(k-1)} + (1 - \alpha) X_{Bk} \tag{14}$$

$$\omega_{Gk} = \alpha \omega_{G(k-1)} + (1 - \alpha) \omega_{Bk} \tag{15}$$

Here, the parameter $\alpha$ controls the balance between long-term and short-term memory of global information, with a larger $\alpha$ favoring long-term memory and a smaller $\alpha$ emphasizing short-term memory. Equation equation 14 describes the fusion of global information accumulated from the first $k - 1$ batches with the local information of the $k$-th batch to construct $X_{Gk}$. Equation equation 15 represents the fusion of global sample weights $\omega_{G(k-1)}$ with the local sample weights $\omega_{Bk}$ to construct $\omega_{Gk}$. $X_{Gk}$ and $\omega_{Gk}$ are then used to optimize the training of the subsequent $(k + 1)$-th batch.

Through cumulative learning and fusion, the weight updates for the current batch become more comprehensive by incorporating information from all previously seen data, thereby achieving the globalization of sample weights.

These weights are incorporated into the DNN's training process by modifying the conventional Mean Squared Error loss function, giving more emphasis to samples that support stable generalization. The original MSE loss is defined as:

$$L_{\text{MSE}}(y, \hat{y}) = \frac{1}{n} \sum_{i=1}^{n} (y_i - \hat{y}_i)^2 \tag{16}$$

where $y_i$ and $\hat{y}_i$ represent the true and predicted values for the $i$-th sample, respectively. Incorporating the learned sample weights $\omega_i$, we revise the loss function as follows:

$$L_{\omega\text{MSE}}(y, \hat{y}; \omega) = \frac{1}{n} \sum_{i=1}^{n} \omega_i (y_i - \hat{y}_i)^2 \tag{17}$$

This reweighted loss allows the DNN to focus more on samples that are less likely to be affected by spurious dependencies, thereby aligning the learning process with more invariant and generalizable patterns. Through iterative training, the stable learning module continuously updates the weights based on cumulative feature statistics, and the DNN adjusts its parameters accordingly, achieving a synergistic balance between predictive accuracy and stability.

# 4 EXPERIMENTS

To evaluate the generalization performance of HSIC-StableNet under distribution shifts, we conduct an experiment using synthetically biased training samples to simulate out-of-distribution scenarios.

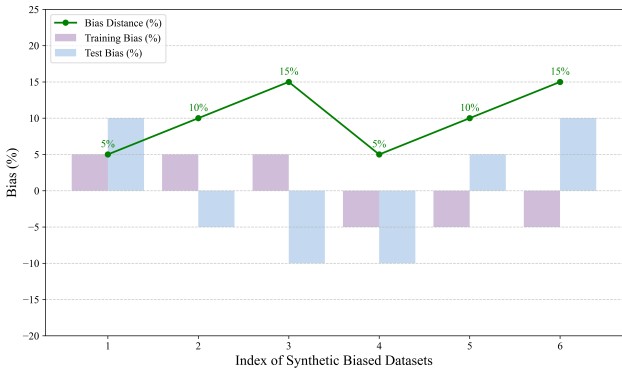

Figure 2: Overview of Synthetic Biased Datasets Constructed Based on $P_5$

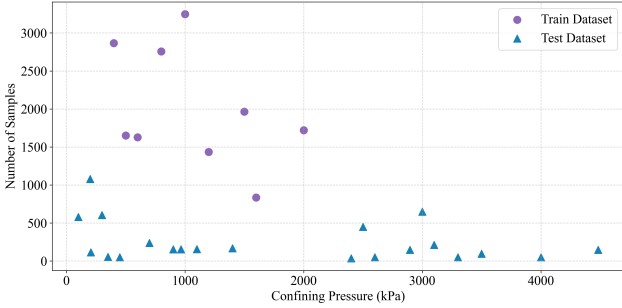

Figure 3: Distribution of training and test datasets by confining pressure

## 4.1 PERFORMANCE EVALUATION

Predicting the deviatoric stress–axial strain ($q$–$\varepsilon_a$) curves of coarse-grained soils is fundamentally a regression task. To evaluate the experimental results, four standard metrics are employed: R-squared ($R^2$), Mean Squared Error (MSE), Mean Absolute Error (MAE), and Mean Absolute Percentage Error (MAPE).

## 4.2 EVALUATING THE GENERALIZATION PERFORMANCE OF HSIC-STABLENET

To assess the generalization capability of the proposed HSIC-StableNet, seven experimental groups were constructed using synthetic biased datasets, generated based on variations in the particle size parameter $P_5$ and confining pressure $\sigma_3$. The first six groups were created by systematically modifying $P_5$, and their distribution characteristics are shown in Figure 2. In this figure, positive histogram values indicate left-skewed distributions, while negative values indicate right-skewed ones. The green curve represents the distributional deviation between the training and test sets for each group.

The seventh biased dataset, constructed based on confining pressure $\sigma_3$, is illustrated in Figure 3, where training samples are marked with purple circles and test samples with blue triangles.

### 4.2.1 EXPERIMENTAL RESULTS ON SYNTHETIC BIASED DATASETS

In HSIC-StableNet, the DNN module comprises four layers, including two hidden layers with 128 and 30 neurons, respectively. The module is trained using a learning rate of 0.0001 and a batch size of 100, with PReLU as the activation function and the Adam optimizer for parameter updates.

To evaluate the effectiveness of HSIC-StableNet, we conduct a comparative analysis against two baseline models: DNN and SNN. The SNN model incorporates a stable learning module that combines the Frobenius norm with an approximate kernel method based on Random Fourier Features (RFF), alongside a standard deep neural network. The DNN components in all models are implemented using the same network architecture and hyperparameter settings.

Table 1: Comparison of $R^2$ across seven synthetic biased datasets

| Model | Index of Synthetic Biased Datasets | | | | | | |
|---|---|---|---|---|---|---|---|
| | 1 | 2 | 3 | 4 | 5 | 6 | 7 |
| DNN | 0.845 | 0.905 | 0.856 | 0.869 | 0.908 | 0.807 | 0.927 |
| SNN | 0.859 | 0.923 | 0.898 | 0.890 | 0.928 | 0.833 | 0.929 |
| Ours | **0.869** | **0.937** | **0.915** | **0.898** | **0.939** | **0.848** | **0.943** |

Table 2: Comparison of MSE across seven synthetic biased datasets

| Model | Index of Synthetic Biased Datasets | | | | | | |
|---|---|---|---|---|---|---|---|
| | 1 | 2 | 3 | 4 | 5 | 6 | 7 |
| DNN | 7.1e5 | 5.2e5 | 8.6e5 | 9.7e5 | 6.2e5 | 9.8e5 | 3.1e6 |
| SNN | 6.5e5 | 4.8e5 | 7.2e5 | 8.7e5 | 4.8e5 | 8.3e5 | 2.9e6 |
| Ours | **6.1e5** | **4.2e5** | **6.4e5** | **7.5e5** | **4.4e5** | **7.8e5** | **2.0e6** |

Table 3: Comparison of MAE across seven synthetic biased datasets

| Model | Index of Synthetic Biased Datasets | | | | | | |
|---|---|---|---|---|---|---|---|
| | 1 | 2 | 3 | 4 | 5 | 6 | 7 |
| DNN | 461.7 | 557.7 | 686.6 | 670.0 | 586.4 | 643.3 | 1044.0 |
| SNN | 447.7 | 530.9 | 606.6 | 631.8 | 523.0 | 582.0 | 1026.0 |
| Ours | **434.2** | **521.2** | **579.7** | **615.8** | **512.7** | **568.6** | **953.0** |

Tables 1–4 summarize the comparative performance of the three models. On the first six datasets, where distribution shifts are introduced based on $P_5$, the proposed model achieves an average improvement of 3.6% in $R^2$ compared to the standard DNN and 1.3% compared to SNN. On Dataset 7, which features distribution shifts based on $\sigma_3$, HSIC-StableNet continues to outperform both baselines, with $R^2$ gains of 1.6% over DNN and 1.4% over SNN.

### 4.2.2 PERFORMANCE ANALYSIS OF THE PROPOSED MODEL

Figure 4 presents a radar chart comparing performance across four metrics for all seven synthetic biased datasets. Datasets 1, 2, and 3 share identical training sets, each exhibiting a 5% left-biased shift based on $P_5$, while their test sets vary, incorporating a 10% left-biased shift, a 5% right-biased shift, and a 10% right-biased shift, respectively. The parameter $v$ represents the distribution deviation between training and test sets.

In Figure 4(a), the green line representing HSIC-StableNet consistently aligns closer to the outer edge than the red (DNN) and purple (SNN) lines, indicating superior $R^2$ performance. Notably, the proposed HSIC-StableNet shows greater improvements as the distribution deviation ($v$) between the

Table 4: Comparison of MAPE across seven synthetic biased datasets

| Model | Index of Synthetic Biased Datasets | | | | | | |
|---|---|---|---|---|---|---|---|
| | 1 | 2 | 3 | 4 | 5 | 6 | 7 |
| DNN | 0.247 | 0.295 | 0.361 | 0.465 | 0.299 | 0.308 | **0.328** |
| SNN | 0.230 | 0.282 | 0.301 | 0.414 | 0.257 | 0.270 | 0.425 |
| Ours | **0.224** | **0.276** | **0.293** | **0.345** | **0.243** | **0.257** | 0.395 |

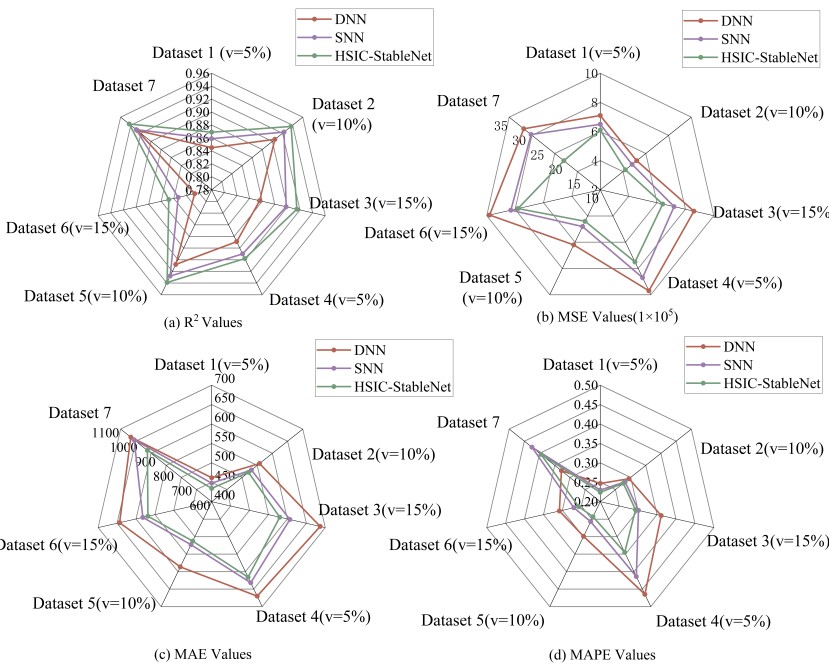

Figure 4: Overview of performance comparisons among HSIC-StableNet, SNN, and DNN models based on synthetic biased datasets.

training and test sets increases, as evidenced by the $R^2$ values across datasets 1, 2, and 3. Additionally, HSIC-StableNet outperforms the baseline methods on dataset 7, which introduces distribution shifts based on $\sigma_3$. These results indicate that HSIC-StableNet consistently maintains robust generalization performance under varying distributional inconsistencies, with its superiority in generalization becoming more pronounced as the degree of distribution deviation increases.

## 5 CONCLUSION

In this paper, we propose HSIC-StableNet, a stable learning framework that combines HSIC-based feature decorrelation with deep neural networks to improve the generalization of strength behavior prediction for coarse-grained soils. Unlike traditional neural networks that struggle under distribution shifts, HSIC-StableNet reduces spurious correlations through sample reweighting. Experimental results show that it outperforms both baseline DNN and existing stable learning methods, achieving robust performance on biased data. Additionally, the model enables accurate prediction for large-particle soils using data from smaller particles, offering an efficient solution to data scarcity in geotechnical applications.

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
