# OpenReview forum: "Robust Strength Behavior Modeling of Coarse-Grained Soils Using HSIC-Guided Stable Learning"
_ICLR.cc/2026/Conference — Submitted to ICLR 2026_

### Official Review · Reviewer_jn9T · 2025-10-24

**Soundness:** 2
**Presentation:** 1
**Contribution:** 1
**Rating:** 0
**Confidence:** 3

**Summary:**

This paper proposes HSIC-StableNet, a stable learning framework that combines exact kernel-based Hilbert-Schmidt Independence Criterion (HSIC) with deep neural networks to predict deviatoric stress-axial strain curves for coarse-grained soils. The method aims to improve out-of-distribution (OOD) generalization by reweighting training samples to reduce feature dependencies.

**Strengths:**

++Addresses a real problem in geotechnical engineering where data collection is expensive and challenging.

++The paper articulates well why OOD generalization is vital for soil strength prediction, given data scarcity for large-particle soils.

+Using exact kernel methods instead of approximations (RFF) for HSIC computation is reasonable for moderate-sized datasets.

**Weaknesses:**

Limited novelty:
A search for “Hilbert-Schmidt Independence Criterion” on Google finds several highly relevant and related papers, such as:
https://arxiv.org/pdf/1910.00270 “We investigate the use of a non-parametric independence measure, the Hilbert-Schmidt Independence Criterion (HSIC), as a loss-function for
learning robust regression and classification models.”
https://proceedings.neurips.cc/paper_files/paper/2007/file/d5cfead94f5350c12c322b5b664544c1-Paper.pdf

Furthermore, a main contribution according to the authors is “While most existing stable learning methods are developed for classification tasks, this work extends the paradigm to regression scenarios by embedding a stable learning mechanism within a regression framework.” However, this paper https://ojs.aaai.org/index.php/AAAI/article/view/6024 from AAAI 2020 considers both classification and regression.


The baselines are very simplistic and not fully described. I would not be able to reproduce the results even if I had the data. However, the datasets are all synthetic, and it is not clear how they are generated.


The tests do not clearly follow the story of targeting soil test strengths. The authors’ synthetic datasets might be modelled after these; however, that is not very clear when reading. The area is significantly outside the domain of ICLR, and should thus be thoroughly explained.


The experiments are not repeated over multiple seeds, even though the training methods are stochastic. There are no error bars, significance tests, or confidence intervals.


Computational cost completely ignored. HSIC computation requires O(N^2) kernel matrix operations. (For each feature pair, the method computes and stores N by N matrices.)
The authors claim this is feasible for "moderate-sized" datasets, but provide no timing comparisons.



The captions are minimal. Please extend these to explain the figures and tables fully.


The related work is almost entirely missing. The section is barely 13 lines… The authors should highlight relevant related work and compare and contrast their method to it.
The authors mention spurious correlations several times, but do not bring this up in the related work.


Missing overview figure. There is a system’s figure in Figure 1, but there is no figure to give an overview of what the authors are doing.


Please align the notation with the formatting instructions.


Grammatical errors:
Equations 1,3,4,5,9,10,13,15, and 17 should all end with a period. The others should end with a comma.


Issues with citations:

“Ma Z. M. Chen Y., Xiong R. When does group invariant learning survive spurious correlations? Advances in Neural Information Processing Systems, 35:7038–7051, 2022” is in the reference list but not cited in the paper.

“coarse-grained soil data are often sparse or imbalanced, especially for large particle sizes, leading to distribution shifts that degrade model generalization” missing citations.

**Questions:**

What do you mean on line 209 with “statistically independent?” What is the exact definition?

“However, since MBGD processes only a subset of samples in each batch, the resulting weights remain localized, which can limit the effectiveness of reweighting in addressing statistical dependencies across the entire dataset.” If each batch took a random subset of the features, why would they remain localized? Please elaborate on this or provide references.

How do you define the bias (%) in Figure 2?

What is seen on the y-axis of Figure 3? I do not understand it.

Why do you only test synthetic datasets? How do you create the synthetic datasets? Do they follow physical laws or something else?

---

### Official Review · Reviewer_SJ2h · 2025-10-29

**Soundness:** 3
**Presentation:** 2
**Contribution:** 2
**Rating:** 4
**Confidence:** 4

**Summary:**

This paper proposes HSIC-StableNet, a stable learning framework that uses the Hilbert-Schmidt Independence Criterion (HSIC) with exact Gaussian kernels to improve out-of-distribution (OOD) generalization in predicting deviatoric stress–axial strain (q–εₐ) curves for coarse-grained soils. The method replaces approximate kernel methods (e.g., Random Fourier Features) with exact kernel computation and extends stable learning—typically used for classification—to regression. Experiments on synthetically biased datasets show improved performance over a standard DNN and a prior stable learning baseline (SNN) across R², MSE, MAE, and MAPE. The authors highlight the model’s ability to predict large-particle soil behavior using only small-particle training data, addressing data scarcity in geotechnical engineering.

**Strengths:**

1. The authors successfully extend stable learning, a concept predominantly used in classification, to the regression domain. This is a convincing and valuable adaptation that broadens the applicability of these methods.
2. The use of exact Gaussian kernels is a well-reasoned choice. For this specific domain with smaller geotechnical datasets, the authors show this yields measurable gains over approximate methods (e.g., SNN with RFF).
3. The model’s ability to predict soil behavior across different particle sizes with limited data directly addresses real-world challenges in the geotechnical field, making it highly relevant for both academia and industry.
4. Synthetic bias effectively simulates distribution shifts. Performance trends (e.g., gains increase with distribution deviation) support robustness claims.

**Weaknesses:**

1. The paper would be strengthened by a deeper theoretical discussion. It's not entirely clear why the exact kernel leads to better generalization for regression, or under what specific conditions we can expect the cross-scale transfer to hold. A discussion of the underlying causal mechanisms (e.g., identifying spurious versus invariant features) is a notable missing piece.
2. The use of synthetically biased data is a good controlled test, but it may not prove robustness against the complex distribution shifts encountered in practice. A real-world transfer experiment (e.g., train on one soil type, test on another geological region) would better validate practical robustness.
3. Given ICLR's focus on core machine learning principles, the highly applied, geotechnical nature of this work might be a less immediate fit. The community might question whether the methodological contribution is foundational enough, or if it is primarily a successful application of existing tools.
4. The experiments compare HSIC-StableNet primarily against DNN and SNN. To strengthen the paper, including comparisons with other state-of-the-art models in OOD generalization or domain adaptation would provide a broader context.
5. The dataset is not public, and key hyperparameters (e.g., kernel bandwidth σ, globalization factor α) are not thoroughly reported. While code sharing is often post-acceptance, these omissions currently limit verifiability.
6. No ablation study isolates the effect of the globalization module (Section 3.3.2). Is it necessary?

**Questions:**

1. What justifies the assumption that decorrelating input features leads to learning invariant mechanisms? Could you clarify which features are considered spurious vs. causal in the soil strength prediction task?
2. How does the model performance scale with much larger datasets in real-world applications, and are there any optimization strategies in place to handle the computational burden of HSIC?
3. How sensitive is performance to the kernel bandwidth σ and globalization factor α? Were these tuned via validation, and if so, how?
4. While exact kernels are feasible here, could your approach be adapted to larger datasets via Nyström approximation or other scalable HSIC estimators without significant performance loss?
5. Can you provide a direct ablation comparing exact HSIC vs. RFF-based HSIC within the same stable learning framework? The current comparison to SNN conflates kernel choice with other architectural differences.

---

### Official Review · Reviewer_tvJM · 2025-10-31

**Soundness:** 3
**Presentation:** 2
**Contribution:** 1
**Rating:** 2
**Confidence:** 4

**Summary:**

This paper proposed a learning framework for improving the robustness and generalization of regression models under distributional shifts.
The application of interest is predicting deviatoric stress–axial strain of coarse-grained soils, where data are scarce and heterogeneous. The key idea is to combine exact kernel-based Hilbert-Schmidt Independence Criterion (HSIC) with deep neural networks to decorrelate features through sample reweighting and weight globalization. This goal is to reduce spurious correlations. Numerical experiments are conducted on synthetic datasets only. The proposed approach is compared to only one baseline, SNN, which it outperforms.

The lack of comparisons to existing domain generalization works, narrow focus on one application, and evaluation on synthetic data only severely limits the contributions of the paper.

**Strengths:**

- The application domain, predicting deviatoric stress–axial strain of coarse-grained soils, is quite interesting. It is an example of a real-world system where data is scarce and heterogeneous, and there is a need for methods that work well for such scenarios.
- From an application perspective, it promises a data-driven alternative to costly triaxial tests.
- There is a consistent and moderate improvement across multiple distribution-shift scenarios, albeit on synthetic datasets.

**Weaknesses:**

- Experiments are conducted on a single domain (soil mechanics), that too only on synthetic biases. It is unclear whether the approach generalizes to other regression tasks or modalities.
- The proposed approach is compared only to one baseline (SNN). There is no comparison the many methods proposed for domain generalization. Further, it is unclear what this SNN baseline is. The acronym is never defined and SNN is never directly cited.
- There are many design choices in the algorithm. But there are no ablation studies to check which ones are sensitive, how to select them, etc.
- The proposed approach has limited novelty. It is able to use the full kernel without approximation due to the limited dataset size. Other approaches which approximate the kernel could also afford to not do the approximation for the same dataset. So, the real contribution of the paper is unclear. Applying to new domain without ML contributions is insufficient for an ML conference.

**Questions:**

- How sensitive is the model to the choice of Gaussian kernel bandwidth?
- How does the model compare to the plethora of existing domain generalization methods?
- Real-data is often noisy in such cases. How robust is the method to noise?
- How would the model perform on real-datasets? Perhaps, it can be trained on synthetic but evaluated on real-data.
- How much benefit does the globalization module provide over local reweighting?

---

### Official Review · Reviewer_QqDp · 2025-11-01

**Soundness:** 2
**Presentation:** 2
**Contribution:** 2
**Rating:** 4
**Confidence:** 5

**Summary:**

This work introduces a stable learning framework based on the Hilbert-Schmidt Independence Criterion to address the distributional shift problem in OOD generalization. In predicting deviatoric stress-axial strain curves that represent the strength characteristics of coarse-grained soils, the model consistently surpasses conventional DNN models and a previously introduced stable learning approach, and demonstrates strong performance in estimating the strength behavior of coarse-grained soils with large particle sizes by utilizing
data samples from soils with smaller particles.

**Strengths:**

1 A stable learning framework based on the Hilbert-Schmidt Independence Criterion to address the distributional shift problem in OOD generalization.
2 Strong performance in estimating the strength behavior of coarse-grained soils with large particle sizes by utilizing
data samples from soils with smaller particles is demonstrated.

**Weaknesses:**

1 The dataset should be explained in more detail.
2 It is not clear how the improvement demonstrated in the new framework matters in addressing the OOD problem of soil mechanics. More discussion on the problem should be discussed.

**Questions:**

1 The accuracy of the present approach is better than others, but marginally. Can you conclude how does this minor improvement help in the problem of soils.
2 If we want to predict large-grain mechanics from small-grain ones, is such a statistical learning approach enough? Can we exclude the physics bias or how can be measure the bounds of generalization? Some practical guidelines should be given.

---

### Official Review · Reviewer_VWVh · 2025-11-01

**Soundness:** 2
**Presentation:** 3
**Contribution:** 1
**Rating:** 2
**Confidence:** 4

**Summary:**

AI models are not robust under distributional shifts in data for this engineering application of analyzing the strength of coarse-grained soils. The paper proposes a solution that reweighs training samples to stabilize a training module which is integrated with a deep neural network. The approach is compared with SNN on several metrics. Their approach calculates the Hilbert-Schmidt Independence Criterion (HSIC) directly rather than using an approximation, because the dataset size allows for it.

**Strengths:**

This paper applies stable learning, a recently introduced method, to a novel problem. The primary contribution of the paper is demonstrating that using the exact calculation of HSIC improves accuracy over the SNN method that uses an approximation (and over the baseline of a neural network without reweighting). Results are demonstrated on synthetic datasets with known amounts of distributional shift; and score using several metrics.

**Weaknesses:**

The primary conclusion is that exact HSIC performs better than an approximation on this one application. This result is not surprising, as it is the more computationally-expensive and exact calculation. The result will only generalize to other problems with similar sizes of datasets, but the size of this dataset is not clear. There is no exploration of dataset sizes.

The result tables show very large errors. What range of values are being regressed? It is likely that there are very large values that are hard to predict. Normalization or quantization may help with this. It would be insightful if the paper showed a scatterplot of at least a subset of the predictions versus targets to better understand the R^2 values.


Minor issues:

It would be helpful to indicate in the caption of Figure 3 that this is dataset index 7.

Fonts in plots are too small to read.

**Questions:**

The paper demonstrates that exact HSIC performs better than an approximation on your data. How would somebody wanting to apply your approach know if it is tractable for their own data? How much data is tractable? There could be some exploration of the computation/accuracy tradeoff to help guide others toward choosing between the approaches.

In Figure 3 and the dataset it is showing, what is confining pressure? It is not the value being regressed. So, is it one of the input features?

What R^2 or error level is needed to make it practically useful to use AI for this application? It seems like the errors will need to drop by orders of magnitude rather than incremental amounts.

---

### Meta-Review · Area_Chair_NMnK · 2026-01-04

**Summary:**

Thanks to authors for submitting their paper to ICLR.

The main concerns below from reviewers are driving the decision to reject the paper at the current form.

1. The reviewers think the paper does not have enough new machine learning ideas. The authors say they use "exact" HSIC because the data is small. But reviewers say this is not a big discovery and it is not surprising that it works better if approximation is used
2. The use of HSIC for regression was used also before as pointed out by reviewer in AAAI'20 work
3. The experiments that use synthetic data. This does not prove the model works in real world.
4. It is hard for other people to do the same experiment because many details is missing.
5. The paper has many small mistakes that make it hard for reading.

**Reviewer Concerns:**

NA - The authors have not provided rebuttal

**Reviewer Scores:**

NA - no rebuttal was provided and ratings have been consistent across all reviewers

---

### Decision · Program_Chairs · 2026-01-26

Reject